# Photoacoustic Imaging Probes Based on Tetrapyrroles and Related Compounds

**DOI:** 10.3390/ijms21093082

**Published:** 2020-04-27

**Authors:** Jean Michel Merkes, Leiming Zhu, Srishti Ballabh Bahukhandi, Magnus Rueping, Fabian Kiessling, Srinivas Banala

**Affiliations:** 1Institute of Organic Chemistry, RWTH Aachen University, Landoltweg 1, 52074 Aachen, Germany; jean.merkes@rwth.rwth-aachen.de (J.M.M.); leiming.zhu@rwth-aachen.de (L.Z.); sbahukhandi@ukaachen.de (S.B.B.); magnus.rueping@rwth-aachen.de (M.R.); 2KAUST Catalysis Centre (KCC), King Abdullah University of Science and Technology, Thuwal 23955-6900, Saudi Arabia; 3Institute for Experimental Molecular Imaging, Forckenbeckstrasse 55, RWTH Aachen, University Clinic, 52074 Aachen, Germany; fkiessling@ukaachen.de

**Keywords:** porphyrins, core-modified porphyrins, phthalocyanines, activatable probes, azacalixphyrin, near-infrared (NIR) dyes, NIR-II dyes

## Abstract

Photoacoustic imaging (PAI) is a rapidly evolving field in molecular imaging that enables imaging in the depths of ultrasound and with the sensitivity of optical modalities. PAI bases on the photoexcitation of a chromophore, which converts the absorbed light into thermal energy, causing an acoustic pressure wave that can be captured with ultrasound transducers, in generating an image. For in vivo imaging, chromophores strongly absorbing in the near-infrared range (NIR; > 680 nm) are required. As tetrapyrroles have a long history in biomedical applications, novel tetrapyrroles and inspired mimics have been pursued as potentially suitable contrast agents for PAI. The goal of this review is to summarize the current state of the art in PAI applications using tetrapyrroles and related macrocycles inspired by it, highlighting those compounds exhibiting strong NIR-absorption. Furthermore, we discuss the current developments of other absorbers for in vivo photoacoustic (PA) applications.

## 1. Introduction

The photoacoustic (PA) effect, discovered by Alexander Graham Bell in 1880, is the generation of sound waves by irradiating an absorber material with light [1]. Using this principle, photoacoustic calorimetric techniques were developed to study the relaxation process of photoexcited chromophores [2,3]. Over the last decade, the application of PA effects was focused on biomedical imaging, in two- and three-dimensions, [4,5] using microscopic [6] and mesoscopic [7,8] modalities. Currently, PA imaging (PAI) is one of the fastest-growing fields in molecular imaging. It allows the use of low energy (< 20 mJ/cm^2^) laser light to generate ultrasound (US) signals, which undergo less scattering and attenuation in tissues than light itself, enable accurate depth encoding, and are detectable using clinical US transducers. This facilitates imaging with the same sensitivity as fluorescence methods (i.e. sub-nanomoles), in the depth and resolution of US (i.e. up to 7 cm and 50 µm, respectively).

Endogenous and exogenous absorbers are used to generate the PA effect. In (pre)clinical practice, irradiation of these agents is carried out with a nanosecond pulsed laser to photoexcite the absorber. Depending on the nature of the absorber, the excited state can convert the optical energy into thermal energy, rising the surrounding temperature by few mK, which then transfers to the neighboring tissue causing thermoelastic expansion generating acoustic pressure waves in a megahertz frequency range. When developing exogenous PA absorbers, it needs to be avoided that the excited states relaxation by light emission (fluorescence) or catalytic production of singlet oxygen from triplet oxygen, thereby causing a loss to the PA generation and potentially toxic effects on the tissue.

Commercial PAI devices enable simultaneous US imaging (B-mode), allowing co-registration of anatomical information, to visualize tissue structures and to locate pathological sites. Three dimensional (3D) PA tomography (PAT) is a commonly used detection, in which several planes of 2D-images are fused into a 3D-image, which still allow the separation of PA signals into respective absorbers; e.g. multispectral optoacoustic tomography (MSOT), where multiple excitation lasers are used in a time-shared fashion to excite the contrast agents and the endogenous background, and spectrally differentiate those signals [9]. With advances in computing power, the PAI devices provide the molecular and functional information on the fly and enable real-time detection [10].

PAI methods are used for in vivo preclinical imaging of a variety of pathological conditions, predominantly in the cancer research, [11,12,13,14] and several clinical translational studies are ongoing [10,15,16,17]. Some indications like, sentinel lymph nodes, [18,19] arthritis, [20] burn wound healing, [21] Duchenne muscular dystrophy, [22] inflammatory-bowl disease, [23] metabolism of brown fat, [24] periodontal pockets, [25] and several other ongoing investigations [26,27,28,29,30] have proven the added value of PAI to the clinical routine.

## 2. PA Devices and Noninvasive Imaging Procedures

Currently, several commercial PA devices are available from different suppliers (Figure 1). These devices are equipped with a light source and ultrasound transducer, using interactive software for fast image analysis. Depending on the optical illumination and acoustic detection configurations, imaging can be performed in 2-D, 3-D tomography, 2-D high resolution microscopy settings and as mesoscopy. The commercial devices, useful for in vivo imaging, are equipped with a nanosecond pulse (Nd:YAG) laser with a frequency of 10 to 20 Hz, which provide pulses with a duration between 7 to 10 ns, and a peak pulse energy of 18 to 26 mJ/cm^2^ in the wavelength range of 680–1100 nm. This energy is within the maximum permissible exposure (MPE) range, which is dependent on the wavelength, pulse frequency and exposure time, e.g. between wavelength of 700 to 1050 nm with a pulse duration of 1 to 100 nanoseconds MPE is 20·10^(2(λ−700)/1000)^ mJ/cm^2^, as defined by the American National Standards Institute (ANSI) [31].

In a typical probe characterization setting, the test solution (in organic or aqueous solvent) is inserted into a small tube (ca 1 mm diameter) and placed in a chamber filled with an intra-lipid solution or water or in vivo mimicking tissues. This solution is irradiated with a laser scanning along predetermined wavelengths, and the generated PA signal is measured using the US-transducer. Often a well-established reference probe is used in same setting to compare the PA signal generation of novel, unknown, probes, to quantify the signal efficiency. In small animal non-invasive imaging protocols, the same laser irradiation and US-transducer procedures are applied, with or without injecting a contrast agent into the animal. Depending on the device set-up, it is possible to either obtain multiple images along a plane, e.g. in whole body imaging (in MSOT), or only a single image targeted at the region of interest like tumors and organs. The signals for endogenous or exogenous probe are then extracted from the measured PA image by spectral unmixing processes.

## 3. Basics of PA Probes: The Known Design Strategies

The efficiency of PA signal generation depends on different factors: a) a fast non-radiative deactivation (internal conversion); b) a higher order excitation (S2 and above) and relaxation; and c) a minimal fluorescence quantum yield are necessary for an efficient contrast agent. Especially the latter property has an inversely proportional effect on the PA signal, as the amount of absorbed energy released by fluorescence is a loss for heat generation (Figure 2, left). Endogenous pigments, such as oxy/dexoy-hemoglobin, melanin, and other materials, such as lipids and collagen, were explored in PA contrast generation, as they exhibit absorption at various wavelengths (Figure 2, right). On the other hand, exogenous PA probes are interesting due to their improved signal to background ratio, targetability, and characteristic spectral signatures. For in vivo imaging, the near-infrared (NIR-I device range: 680–970 nm, NIR-II: 1064 nm) absorbing materials are preferred, as NIR-light undergoes less scattering and absorption in tissue. Therefore, developments of such dyes are particularly in the focus of research [32,33,34,35].

Here, several probe development strategies have been explored, ranging from inorganic plasmonic nanoparticles [36] to semi-conducting organic nanoparticles [33] to organic dyes based on the different cores like polymethines [37,38,39] and black hole quencher dyes [40] as well as many others [41,42,43,44,45,46]. Gold and other noble metals have been studied for very long time; however, due to toxicological or pharmacological concerns as well as costs, it is very difficult to develop them as contrast agents for clinical use. Indocyanine green (ICG), a US Food and Drug Administration–approved polymethine dye with low toxicity and strong NIR-absorption, is the most widely studied compound in PAI. Similarly, other polymethine-based cyanine dyes and cyanine-containing quencher dyes were also explored. Many azo-bridged black hole quencher dyes provide PA activity but possess high toxicity, and thus are not useful for diagnostic in vivo applications [47]. Also, squaraine dyes [48,49] have shown strong optical absorption and photostability, but other properties such as insolubility, aggregation and chemical reactivity (to nucleophilic substitution), have limited their application. In parallel, tetrapyrrole core dyes have been explored, as they exhibit strong absorption and long known biomedical use as therapeutics. 

This review focusses on compounds based on tetrapyrrole or that are inspired by tetrapyrrole. 

## 4. The Tetrapyrroles

Tetrapyrrole chemistry has been attracting attention of scientists for over a century due to their role as “Pigments of Life” and applications in broad areas of science. A major research area of the porphyrinoids is dominated by their therapeutic application as photodynamic therapeutics, e.g. to treat cancerous lesions [50,51,52,53]. More recently, the tetrapyrroles have also been used in other areas such as photodynamic inactivation of multi-resistant bacteria [54,55]. In addition, porphyrinoids are increasingly explored in material applications [56], light harvesting [57,58,59], molecular sensing devices [60], and (photo)catalysis [61,62]. This continued interest stems from their strong absorption of UV-visible light, accompanied by fluorescence emission and formation of triplet state resulting in singlet oxygen. Typically, porphyrins exhibit a very strong absorption band (Soret) in the visible range (<430 nm) and a few weak bands (Q-) in the red to NIR-rage (<700 nm). Therefore, to make tetrapyrroles suitable for in vivo PAI, the following steps are required: a) redshifting the Q-bands into the NIR, b) minimizing the radiative-decay yields, and c) tuning the internal conversion of excited state. To achieve these goals, an appealing route is core-modification, which has been explored to redshift the absorption and to quench the fluorescence emission. 

### 4.1. Natural Tetrapyrroles (Oxy/Deoxy Heme) as PA Probes

An often used endogenous compound in PA applications is the tetrapyrrole-based Fe(II)/Fe(III)protoporphyrin (PpIX) in hemoglobin (Figure 3, top left). The Fe(II)/Fe(III) PpIX exhibits distinct differences in absorbance for oxygen-unbound and -bound forms (Hb, HbO_2_; 750 nm vs. 850 nm, isosbestic point at 798 nm) (Figure 3, top right) and shows detectable PA signals in the red to NIR range. The whole blood PA measurement first gives an overlaid spectrum of both, HbO_2_ and Hb, with respective intensities. By using a mathematical model, the device’s integrated software takes the pre-calibrated 100% HbO_2_ and Hb values, and separates the observed spectrum into respective HbO_2_ and Hb agents. The respective PA signal intensities can be used for the determination of the total hemin concentration, for the determination of the partial oxygen (saturation), and for the identification of hypoxic regions and the heterogeneity in the vascularity of organs and tumors (Figure 3, bottom) [63]. Commercially available PA devices are already equipped with the pre-loaded library of PA spectra of 100% oxygenated (HbO_2_) and deoxygenated (Hb) blood, along with the other commonly used endogenous or exogenous pigments, simplifying their application for non-experienced users.

It is important to note that the practicing physician certainly prefers such endogenous absorbers over injectable probes for characterizing pathologies [64,65] if they can provide the same information. Unfortunately, many important molecular disease markers cannot be sensitively and specifically discriminated by PAI making exogenous PAI probes necessary.

### 4.2. Core-Modified Tetrapyrroles as PA Probes

One of the first tetrapyrrole-based probes for PAI was based on a bacteriochlorin photodynamic therapeutic (PDT) agent, for a combined PAI/PDT approach [66]. Simões and coworkers prepared *meso*-tetra (2,5-dihalogenated phenyl) porphyrin (**1-2H**) with sulfonic acid or sulfonamide groups on the phenyl rings to improve its aqueous solubility (Scheme 1). The core tetrapyrrole was doubly reduced (2,3,12,13-tetrahydroprophyrin) to obtain the bacteriochlorin pigment **2-2H**, which exhibited a strong NIR-absorption band (λ=749 nm; ε = 10^5^ M^−1^cm^−1^). Then, they studied the energy requirement for PDT and PAI, and found that with only 33% of the energy needed for PDT could produce strong PA signal. With this optimization, they performed PA studies of in vitro photoacoustic calorimetry with 5 mJ and minimized phototoxicity, and studied the generated photoacoustic wave features of the probe. In vivo small animal studies were carried out, and comparative studies showed that the dye distribution was primarily concentrated in the liver in first 24 h. Though **2-2H** is a polar and aqueous soluble small molecule dye, due to the function of liver as one of the main clearing organs of exogenous molecules with abundant transporters and fenestrated endothelium pores, large portion of **2-2H** was cleared by liver in addition to kidneys and spleen in >72 h [67].

For solely diagnostic applications, it is preferable to avoid any light-mediated toxicity, such as generation of reactive oxygen species (ROS; e.g. ^1^O_2_). As the porphyrins and/or reduced porphyrins (bacteriochlorins) produce ^1^O_2_ upon excitation, it is required to inhibit such formation. Therefore, the optical properties of the tetrapyrroles were modified. One such route was explored by Brueckner, Zhu, and coworkers via core oxidation (β, β’-oxoporphyrin) and further β-oxo *meso*-annulation methods [68]. Here, they prepared a derivative, β-hydroxylamino oxoporphyrin and fused it with the *meso*-phenyl to obtain a quinoline annulated tetrapyrrole (**3a-2H**, Figure 4), which exhibited a moderate NIR-absorption band (λ=764 nm; ε < 20,000 M^−1^cm^−1^), negligible fluorescence emission, and a very low ^1^O_2_ production. In vitro studies proved that the PA-signal generation efficiency of **3a**-**2H** was over 2.4 times higher than that of ICG. Having found a PA suitable dye, the group then prepared an aqueous soluble quinoline-annulated porphyrin **3b-2H**, by attaching dodeca-ethylene glycol (MeO-PEG-12-OH) at *para*-position of the *meso-*aryl group. This water-soluble dye was then injected in to tumor-bearing mice for accumulation studies, and ICG was used in same setting for comparison. It was found that **3b-2H** exhibited high in vivo tumor accumulation efficiency (Figure 4, right), with about 4-fold higher than ICG within 10 min postinjection, and persistence of this high PA signal for up to 45 min [69]. 

Alternatively, Banala et al. studied a fluorescence-quenched tetrapyrrole, prepared by annulation of four quinones at β, β’-positions of porphyrin, a so called ‘black’ porphyrin (**4**; Figure 5) [70,71] for PAI. By virtue of this extended conjugation and annulation, the black dyes exhibited NIR-absorption (λ=732 nm; ε = 54000 M^−1^cm^−1^, for the Zn^(II)^ insertion) and complete fluorescence quenching [72]. By in vitro phantom analysis, it was proved that **4-Zn** exhibited a 3.2 times higher PA signals than ICG and tissue analysis showed that concentrations of around 1 nmol/mm^3^ were detectable. Using a PEG-300 formulation in phosphate buffered saline (PBS) and injecting the probe into the tail vein of healthy mice, its diagnostic capability was demonstrated. After 1 h circulation time, the mice were sacrificed and ex vivo analyses demonstrated that the dye was mostly accumulated in liver, which is the main clearing organ for hydrophobic porphyrins. 

Zheng and co-workers have approached an elegant route to apply the (reduced) tetrapyrroles for PAI and even for multi-modal imaging by incorporating porphyrin in a lipid-shell microbubble (MBs) containing C_3_F_8_ or C_4_F_10_ or SF_6_ gaseous cores (porphyrin-shell bubbles, Figure 6). Initially, they attached a phospholipid side chain to a chlorin pigment via the C17-propionic acid, which was then self-assembled to a J-J-aggregated microbubble shell [73]. These porphyrin-shell bubbles exhibited a red-shift in absorption along with fluorescence quenching in the intact shells and generated a strong PA as well as US signal by virtue of the incorporated heavy gas in the core of the formulation. The porphyrin-shell microbubble could be easily destroyed with an ultrasound pulse and fell apart to form small nanoparticles, which were also visible in PAI. These nanoparticles circulated for longer periods of time and were internalized by cells [74,75]. Upon internalization, the nanoparticles were disassembled completely and restored the original properties of tetrapyrrole (chlorin) and can be used for both fluorescence imaging and PDT treatment [76,77]. The Zheng group has been expanding this approach to dynamically ‘blacken’ many other dyes for PAI, and in vivo regeneration of the original dye for multimodal imaging applications [78]. Other groups explored the chlorin’s PA applications by tuning its lipophilicity or pH-triggered supramolecular structural changes. For the latter purpose, tricarboxylato chlorin (Cp6) was studied in the pH-induced formation of nanoaggregates and thus generated PA gain for intratumoral pH detection [79]. In another study, a chlorin-poly-arginine conjugate was prepared, which undergoes self-assembly in a polarity dependent manner, thereby giving a redshifted PA signal [80]. Further, the system was used to quantify enzymatic activity in vivo, by attaching chlorin to an enzyme-cleavable peptide chain, using the aggregation induced shift and gain in PA signal [81].

Instead of modifying the porphyrin core, Wanli et al. performed periphery modification of porphyrin in a donor-acceptor approach to tune the electronic and optical properties [82]. 

It is known that the conjugation of strongly electron-withdrawing tetracyanoethene (TCNE) or tetracyanoquinodimethane (TCNQ) groups with electron rich anilines via a π-system leads to NIR-absorption [83]. Wanli et al. exchanged the π-system with a porphyrin, that contained a 4-(*p*-alkynyl-anilinyl) phenyl at *meso* position (**5**) to tune the absorption [82]. In this work, the authors systematically changed acceptor electron-withdrawing efficiency of tetracyanodimethanes with the porphyrin donor-π system (**6a-c**, Scheme 2). This insertion reaction first underwent a (2 + 2) cycloaddition with the alkyne followed by a ring opening sequence yielding a series of dyads with embedding porphyrin, which exhibited strong NIR absorption. Subsequently, photophysical and PA measurements of these porphyrins were carried out, and it was found that the double incorporation of highly electron withdrawing tetrafluoro-tetracyanoquinodimethane (F_4_-TCNQ) lead to an intense NIR band (λ = 866 nm; ε = 5.6 × 10^4^ M^−1^cm^−1^) and stronger PA signal than the other dyads. Although this porphyrin incorporation in donor-acceptor dyad redshifted the charge-transfer absorption band, the relative increase in size, and hydrophobicity demand redesign of such acceptor incorporation systems for biomedical applications.

In addition to the stand-alone use of core modified porphyrins as contrast agents, Banala et al. have explored core modified porphyrins towards detection of short-lived reactive oxygen species *via* trigger-induced PA signal generation (Figure 7). Previously, for ROS detection, fluorogenic strategies were mostly pursued [84,85]. Here it was attempted to detect superoxide ions (O_2_^−^) by PA by synthesizing a porphyrin with four *meso*-butylated *p*-hydroxy phenyl (BHT unit), that upon oxidation formed a quinomethide fused porphyrin (oxoporphyrinogen; OxP, **8**). The OxP exhibited completely different optical properties and reactivity than porphyrins. The pristine OxP did not exhibit a NIR-absorption (λ = 654 nm); however, upon capturing the O_2_^−^, the absorption maximum was redshifted by up to 240 nm and showed an over 10-fold increase in the PA signal (along with the increase in molar absorption coefficient) [86]. Among the variety of ROS and reactive nitrogen species (RNS) explored in vitro, PAI studies showed high selectivity and maximum signal gain for O_2_^−^ (Figure 7c).

As the ROS play an important role in biology, non-porphyrin chromophores have been employed in PA detection approaches, which produce ROS-activated gain in PA signals or redshifted PA signal maxima (towards NIR) [87,88]. 

## 5. PA Applications of Phthalocyanine (Pc) and Napthalocyanine (Nc)

The phthalocyanine (Pc) macrocylic core resembles the tetrapyrrole, albeit the pyrroles are fused with butadienes (i.e. isoindoles in the core) and napthalocyanine (Nc) via benzodiene. The Pcs are the structural mimics of tetrabenzoporphyrin dyes substituted with four aza-groups at meso-positions. By virtue of the extended conjugation and electron rich nitrogen-atom at *meso*-position, Pcs and Ncs absorb strongly in the NIR-range exhibiting ε > 3x10^5^ M^−1^cm^−1^ and have been thoroughly studied for a wide range of applications [89]. The central core of Pcs and Ncs can accommodate a variety of metals and non-metals, which have been explored for tuning the optical properties and photodynamic activity [53,90]. By virtue of high photostability and large molar absorbance, Pcs and NPcs have attracted attention as molecular PAI probes [91] and were explored by several groups for the purpose of biomedical imaging. Some interesting PAI and theranostic applications will be discussed in the following:

Olivo and coworkers explored the suitability of tetrasulfonic acid-functionalized phthalocyanines (**9**, PcS4), as metal-free (**9a**, H2-PcS4), Zn(II)-inserted (**9b**, ZnPcS4) and Al(III)-inserted (**9c**, AlPcS4) forms for PAI (Figure 8), in phantoms and in tumor-bearing mice [92]. These PcS4s have shown absorption and PA maxima close or below 680 nm, and a tail band in the NIR. PAI experiments with linear dilutions indicated that H2-PcS4 exhibit highest contrast, and good spectral overlap with its absorption spectra, followed by Zn-PcS4, for concentrations from sub nmol/mL to 10 nmol/mL. The least contrast was generated by AlPcS4 with an about 10× lower signal than H2-PcS4. In bio-distribution studies, H2-PcS4 showed higher tumor accumulation than the other two, 1 h post administration, suggesting that the central metal substantially influences the pharmacokinetic properties of the Pcs. 

Ntziachristos and coworkers investigated the suitability of Si-inserted Nc (**10**, Figure 8) for in vitro and in vivo theranositc use, as it exhibits a strong NIR-absorption (λ = 770 nm; ε > 5 x 10^5^ M^−1^cm^−1^), fluorescence and ^1^O_2_ generation [93]. As the SiNc is known for its high hydrophobicity, a cremophore EL formulation was used to increase the aqueous solubility. They found in power-dependent measurements that the required light energy to produce a PAI signal was far lower than generation of ^1^O_2_. From this characterization, it was proposed to use SiNc as light-dependent tunable contrast agent and therapeutic probe. However, there was a limitation in the probe that the exhibited PA spectrum was not matching to the observed absorption spectrum (in phantom), and that probe’s tendency to aggregate complicated spectral unmixing. To circumvent this, Lovell and coworkers explored an α-oxygenated Nc (ONc) using Sn^(IV)^–metalation (**11**), for which the axial coordination site provided extra handle for the attachment of PEG-10K / PEG-20K to improve aqueous solubility. The mono-PEGylated ClSnONc (**11**, SnONc) exhibited redshift absorption at 930 nm, suitable for deep tissue imaging. The authors prepared a polysorbate formulation of the PEG-SnONc for in vivo application and found up to 10 times longer circulation by virtue of stable PEG-attachment, compared to non-PEGylated ONc derivative [94].

Since the central metal ion has profound impact on PAI signal generation, a systematic investigation of metal effects in napthalocyanines (Figure 9) was carried out by Strassert and coworkers [95]. They looked at four similar Ncs (**12**, **13**) varying only in central ions (CuNc, NiNc, VNc, SiNc **10**). The metallic centers with unfilled valence electron configuration (Cu^2+^: d^9^; Ni^2+^: d^8^; VO^2+^: d^1^) promoted a fast radiationless deactivation generating heat and produced a stronger PAI signal compared to the empty valence shell configuration (Si^4+^: d^0^). Their optical and photoacoustic properties were quantified in toluene and cremophore EL, and compared with two broadly used cyanine fluorophores (IR 800 CW dye and Cy7) under identical conditions. The Cy7 and IRDye exhibited a significant photobleaching than Ncs under irradiation. They showed that the highly fluorescent SiNc **10** was a weak PA probe than nonfluorescent Ncs; CuNc and VNc were aggregated, as identified by shoulders of their Q-bands, and converted less excitation energy to thermal energy. Highly soluble but non-aggregating NiNc performed best, produced the highest PA signal, and exhibited also no phototoxicity (^1^O_2_). Therefore, from these studies they could prove that, in the development of photoacoustic probes suitable for in vivo imaging, one has to consider influence of variety of factors including core and periphery. 

Along that line, Kobayashi et al explored the influence of strongly electron withdrawing non-metal central atoms [96]. They quantified computationally that high valence-small ionic radii ions in the core, and electron rich substitutions at the α-position of Pc periphery, can bring the valence molecular orbitals closer. With a phosphorus atom (as P^5+^; PPcs) in the core and the electron rich thiophenyl at the periphery, the absorption maximum of PPcs could be shifted up to 1018 nm (ε = 4 × 10^4^ M^−1^cm^−1^). Lovell and coworkers used this design principle and prepared a similar PPc that exhibited a λ_max_ 1042 nm (ε = 1.1 × 10^5^ M^−1^cm^−1^ in CHCl_3_) for PA applications [97]. Since the PPc is highly hydrophobic for biomedical imaging applications, it needs to be formulated in surfactants (Tween, Pluronic etc.) to achieve aqueous solubility. This could be achieved using Tween 80, resulting in a formulation with absorption at λ_max_ 997 nm. For PAI, the authors used a broadly accessible 1064 nm laser (Nd:YAG pulse). The PPc formulations were explored in tomographic imaging at high depth PAI by trans-illuminating the dye solution through a chicken breast layer and a healthy human limb to successfully detect up to 11.6 cm and 5 cm deep, respectively. Furthermore, passive accumulation of the PPc formulation in tumors was studied after intravenous injection in mice, and intestinal transport function was assessed after oral administration.

Although excellent PA signals were obtained from existing Pcs and Ncs, their insolubility in water hinders their biomedical use. Several formulations were investigated to overcome this obstacle. Prud’homme and coworkers explored flash nanoformulations (FNP) using block copolymer directed self-assembly and surface coatings with polyethylene glycol (PEG) [98]. Upon the incorporation of the (N)Pc dyes, they obtained water-dispersible nanoparticles with sizes from 38 to 88 nm, whose absorption were in the NIR range. Furthermore, they showed that the PEG-arm on the surface could be used for functionalization, e.g., for folate receptor targeting. Lovell and coworkers used a different approach for loading Pcs and Ncs into micelles. For this pluronic, F-127 was used, which shows a concentration-sensitive critical micellar temperature and can reversibly be switched to incorporate hydrophobic (N)Pcs in nanoparticulate to microcrystalline form. The Lovell group utilized this property to obtain a minimal surfactant-contained colloidal suspension. For this purpose, a standard F-127 micellar formulation was prepared first, which when cooled to 4 °C, and maximal amount surfactant was stripped off, forcing the dye molecules to form nano aggregates with minimal surfactant (Figure 10) [99,100]. Then, the free and loose surfactant were removed by centrifugal membrane filtration at 4 °C, producing Nc colloidal suspension, which was called “frozen micelles” by the authors. Interestingly, the process did not alter dyes’ absorptions and showed suitability for NIR-II range PAI (1064 nm). Furthermore, the frozen micelles were found to be highly stable at the low pH values present in the stomach. The micelles passed safely through the intestine without any adsorption upon oral insertion and thus allowed imaging of the dynamics of the gut passage by PAI in real-time [99]. 

In addition to standalone applications of Pcs and Ncs for PAI, trigger-responsive probes were explored, in particular, for the detection of short-lived reactive oxygen species (ROS). Pu and coworkers designed a Pc-probe based on the conversion of a hydrophilic to a hydrophobic dye, combined with a self-assembling nanosystem [101]. They prepared a ZnPc probe containing four H_2_O_2_-cleavable boronate synthons linked to a self-immolative spacers with poly(ethylene glycol), mPEG-45. By virtue of four PEG units, the whole construct was water soluble and formed a monomeric unit that gave a weak PA signal similar to that of standard ZnPc. Upon the reaction with ROS, the boronate became a phenol and initiated spontaneous self-immolation to cleave off the linker and mPEG-45, generating hydrophobic ZnPc, that underwent the self-assembly process. This formed ZnPc nanoparticles, which showed an increased rate of heat transfer yielding enhanced PA signals. The use of thie ROS-activatable PEG-ZnPc was further characterized in vivo using glutathione inhibitors that allowed the accumulation of H_2_O_2_ and proposed it as a marker for quantifying ROS production during chemotherapy.

Uchiyama and coworkers approached the ROS detection with a novel design inspired by Pc, namely benziphthalocyanine (BPc, **14c**), incorporating a phenol in the Pc macrocycle (Figure 11) [102]. The BPc exhibited a quinoid-phenolic equilibrium and a high photostability, as Pcs, but weak NIR absorption 870 nm (ε = 1.64 × 10^3^ M^−1^cm^−1^) and no fluorescence emission. The equilibrium of **14c** could be blocked by methoxylation, which shifted λ_max_ from 870 to 584 nm (ε = 5.31 × 10^3^ M^−1^cm^−1^). This proved that the presence of a free hydroxyl group was essential for tautomerism in BPcs. The Uchiyama group used this principle and decorated the BPc with thiocarbonate (**14a**) and acetoxybutanoylate (**14b**) as esterase or H_2_O_2_ cleavable groups, respectively, to generate trigger-responsive probes for PAI. Although these reactive BPcs did not give a large gain (<5×) in PA signal, their activation could be detected by a large redshift in PA maximum (>300 nm, into NIR). This unique feature change in conjugation in response to in vivo triggers could be further explored in NPc-based designs for NIR-II imaging.

## 6. PA Probe Development in NIR-II Range

There is a high demand for PAI probes exhibiting absorption in the second biological transparency window (NIR-II: 1000 to 1400 nm) as indicated in earlier sections. Current probe research has, so far, not utilized the porphyrin or phthalocyanines (except PPc) for PAI in the NIR-II range. There was a limitation in the porphyrin or Pcs to achieve the NIR-II absorption due to constraint in the core electronic configuration. Thus, to overcome this limitation, recently, tetrapyrrole-inspired tetraaza tetrabenziporphyrin (azacalixphyrins) was designed, which contained 18-π electrons in the macrocycle system [103]. This design completely avoided the pyrrole and used tetraiminobenzene in macrocyclization, thus displaying a bis-zwitterionic character: anionic core and cationic periphery (**15**, Figure 12). The azacalixphyrins exhibited a NIR-absorption band (λ_max_ > 880 nm, ε > 10,000 M^−1^cm^−1^), which can be redshifted by substituting imines with electron-rich aryl groups up to λ_max_ of 941 nm with ε > 20,000 M^−1^cm^−1^. It was found that the PA spectra directly correlate with the absorption spectra, and a linear dependency of concentration to signal strength was observed [104]. Similarly, a dimeric fused bis-azacalixphyrin (**16**), sharing an aromatic unit, was reported recently, which exhibited a strong absorption band in the NIR-II window (λ_max_ = 1029 nm, ε = 17,800 M^−1^cm^−1^) [105]. This bis-fused **16** could be further fine-tuned, and PA applications are yet to be reported. 

Previously, NIR-II absorbing (λ_max_ > 1050 nm) bis-fused porphyrin and fused (poly) porphyrins have been reported [106,107]. However, such multi-porphyrins tend to form aggregates due their high hydrophobicity, which is detrimental for in vivo PAI applications and thus not yet applied in imaging. To overcome this, an elegant design was recently presented by Furuta, Kim, and coworkers, which was based on a dioxohexaphyrin, as NIR-II-light responsive dye [108]. The dioxohexaphyrin contained 6-pyrroles in the core and showed *N*-confusion, giving NNNN/NNOO and NNNO/NNNO ligands for metal (M = Zn and Cu) complexation, giving two electronic configurations (28 π and 26 π). The metalation yielded geometrical isomer, cisoid or transoid (cis **17a-M_2_**, trans-**17b-M_2_**; Figure 13a,b), exhibiting geometry-dependent photophysical properties. The cisoid exhibited NIR-II tail absorption and a weak emission without a PA response (Figure 13c,d), whereas the transoid exhibited an intense band > 1010 nm, a stronger PA signal (at λ = 1064 nm, up to 1.4-fold) than the cisoid isomer. This was a first step in using expanded porphyrins as NIR-II dyes for PAI, which we believe will stimulate research on fine-tuning the properties and electronic configuration of such macrocycles.

## 7. Conclusions

Here, we summarize the current developments in PAI probe design based on tetrapyrroles and inspired compounds. Though we present promising starting points to understand the influence of photophysical properties on PAI signal generation, there is still a lot of fine-tuning necessary. Foremost is the shifting of the weak Q-band absorption of porphyrin (λ_max_ < 700 nm, ε = 5000 M^−1^cm^−1^) to the NIR. Here, the approach explored by Brueckner et al., the quinolino porphyrin, can serve as a model to change the core electronic configuration of the macrocycle. This suggests that a complete redesign of the porphyrin is necessary to generate strong Q-bands. In such a redesign approach, we think it will be highly appropriate to explore heme as a biocompatible, modifiable synthon. Especially, designing quencher compounds like heme-inspired verteporfin could possibly yield a PA-suitable biocompatible small molecule probe. If a pyrrole modification in verteporfin results in the absorption in the clinically relevant range (λ_max_: 682 nm), why can it not be exploited beyond to prepare PA suitable quenchers?

Another growing field demanding new probes is NIR-II-PAI, as the 1064 nm laser (Nd:YAG pulse) is now broadly accessible and allows imaging in more than 10 cm depths in tissue [109]. So far, only a few compounds exhibiting this range are available. Examples are dioxohexaphyrin, PPc, and ONPc, which could serve as lead compounds for future designs. However, fine-tuning the PPc and NPc will be challenging, as they are highly hydrophobic and not amenable for optimization of physicochemical properties. On the other hand, it will be interesting to see what other smaller ions than P^5+^ could be inserted in Pcs and whether NPcs core could accommodate P^5+^ and their absorption maxima. We could expect to explore non-tetrapyrrolic cores for this purpose and designs based on benziphthalocyanine (BPc). Other designs based on benzobis(thiadiazole) (BBT), a donor–acceptor–donor conjugate, is very attractive as the acceptor. So far, as donors, triarylamine or thiophenes have been widely studied with BBT in NIR-II fluorescent and NIR-I PA imaging [33]. Although the currently known BBT probes still show their λ_max_ < 900 nm, computational studies have shown that the BBT-inspired core as acceptor is suitable for tuning the absorption to the NIR-II range [110]. Other probes, inspired by the cyanine-core, have been explored [39,111]; however, the photostability of polymethine bridged compounds still poses a critical limitation for PAI. As macrocyclic conjugated systems exhibit high photostability, with or without a pyrrolic core, as in azacalixphyrin or benziphthalocyanine, they could further be explored in the field of NIR-II probes. In this context, as the chemical synthesis is a time-consuming process, computational predictions should be used to choose the right targets.

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
