# Peer review of "Photoacoustic Imaging Probes Based on Tetrapyrroles and Related Compounds"

_ijms, 2020, doi:10.3390/ijms21093082_

Round 1

Reviewer 1 Report

Interesting report.

Some points.

  1. Also express (< 20 mJ/cm2) in the form of what wavelength radiation in nm/sec would be more useful for readers to understand what type of laser radiation would work.
  2. “Endogenous pigments, such as oxy/dexoy-hemoglobin, melanin, and other materials, such as lipids, and collagen were explored in PA contrast generation, as they exhibit absorption at various wavelengths (Figure 1, right).” What consequence does it have on the application of PA to people of color, specifically black skinned people? Would this not severely limit the extent of this application?
  3. “The Fe(II)/Fe(III) PpIX exhibits distinct differences in absorbance for oxygen-unbound and -bound forms (Hb, HbO2; 755 nm vs. 700 nm, isosbestic point at 798 nm) (Fig 2 right).” Should this be 755 vs 900 nm? Can’t see a peak at 700. Some explanation as to how the PA signal (a relaxation) is identical to the absorption would also be useful here. Is there no stokes shift with PA? Are the peaks for the PA material in the sound wavelength region?
  4. “The total PA signal intensities can be used for the determination of the total hemin concentration and the ratio in the PA signal for the determination of the partial oxygen (saturation), and thus for the identification of hypoxic regions and the heterogeneity in the vascularity of organs and tumors.” Exactly how would this be accomplished? I cannot appreciate how this technique would provide 3-D information. Some images here would be useful.
  5. Figure 4 A and B show exactly what part of the mouse? What does the color scale and the part of the image in color signify?
  6. Rest of the paper summarizes various applications and looked like abstracts/conclusions from the publications.

The review would be greatly enhanced if at the beginning an explanation of the various techniques to utilize PAI was mentioned, i.e, how is this experiment conducted and what sort of equipment is required. Then some statement as to the proposed applicability followed by the present contents would render this far more useful.

Author Response

Reviewer 1
Comments and Suggestions for Authors

Interesting report.

Thank you.

Some points.

1. Also express (< 20 mJ/cm2) in the form of what wavelength radiation in nm/sec would be
more useful for readers to understand what type of laser radiation would work.
This information is now included in the revised draft
Typical commercial devices are equipped with a nanosecond pulse (Nd:YAG)
laser with a frequency of 10 Hz to 20 Hz, which provide pulses of duration
between 7 to 10 ns, with peak pulse energy of 18-26 mJ/cm2 in the wavelength
range of 680 nm – 1100 nm. This energy is within the maximum permissible
exposure (MPE) ranges, which depends on the wavelength, pulse and exposure
duration, e.g. wavelength, 700 nm to 1050 nm with pulse duration 1 to 100
nanoseconds, MPE is 20*10(2(λ−700)/1000) mJ/cm2, as defined by the American
National Standards Institute (ANSI). The light sources which produce this energy
range (or lower) works for PA imaging.

2. “Endogenous pigments, such as oxy/dexoy-hemoglobin, melanin, and other materials,
such as lipids, and collagen were explored in PA contrast generation, as they exhibit
absorption at various wavelengths (Figure 1, right).” What consequence does it have on
the application of PA to people of color, specifically black skinned people? Would this
not severely limit the extent of this application?
This is true; the endogenous PA signals, including the melanin (colour of skin)
can affect the quality of the data and image. However, by applying a proper
mathematical correction (adjusting for higher % of melanin) and spectral
delineation of the endogenous pigments, the exogenous probe generated signal to
noise ratio can be increased.

3a. “The Fe(II)/Fe(III) PpIX exhibits distinct differences in absorbance for oxygen-unbound
and -bound forms (Hb, HbO2; 755 nm vs. 700 nm, isosbestic point at 798 nm) (Fig 2 right).”
Should this be 755 vs 900 nm? Can’t see a peak at 700.
Sorry for a mistake here; Hb and HbO2 produce their PA maxima at 750 nm and
850 nm, respectively.
The whole blood PA measurement first gives an overlaid spectrum of both, HbO2
and Hb, with respective intensities. By using a mathematical model, the system’s
integrated software takes the pre-calibrated 100% HbO2 and Hb values, and
separates the observed spectrum into respective HbO2, Hb agents.

3b. Some explanations as to how the PA signal (a relaxation) is identical to the absorption
would also be useful here. Is there no stokes shift with PA? Are the peaks for the PA material
in the sound wavelength region?
The produced PA spectrum highly depends on the probe properties: if internal
conversion from S1-excited state occurs, then the PA spectrum is identical to the
absorption, which is often observed for metallated tetrapyrroles compounds.
However, a shift in PA maximum over absorption was observed for example in
BODIPY and squaraine chromophores.
The produced sound waves are in the range of 70 dB, which are captured with 13
to 55 MHz transducer (in preclinical research), at all excitation wavelengths.

4. “The total PA signal intensities can be used for the determination of the total hemin
concentration and the ratio in the PA signal for the determination of the partial oxygen
(saturation), and thus for the identification of hypoxic regions and the heterogeneity in the
vascularity of organs and tumors.” Exactly how would this be accomplished? I cannot
appreciate how this technique would provide 3-D information. Some images here would
be useful.
The 3D-information is obtained by the following: multiple images are captured
from different planes using a 2D-transducer connected to a 3D motor, and finally
fusing of all images will produce a 3D-image (similar techniques are used in
computed tomography (CT) and 3D-clinical ultrasound imaging). From this 3D
image, a respective region of interest is chosen to extract the signals related to
endogenous and exogenous pigments.
A few example images, of 2D and 3D-PAI are now included in section 2 (as
Figure 3).

5. Figure 4 A and B show exactly what part of the mouse? What does the color scale and the
part of the image in color signify?
The parts of the mouse shown in images A and B are now explained. The colour
scale is in arbitrary units (a.u.), typically the highest is dark red. This has been
clarified.

6. Rest of the paper summarizes various applications and looked like abstracts/conclusions
from the publications.
We attempted to condense text passages, giving a glimpse of the importance to
chemical design and achieved imaging efficiency, as well as discussing theobservations for further tuning. Now, we elaborate the explanation style.

7. The review would be greatly enhanced if at the beginning an explanation of the various
techniques to utilize PAI was mentioned, i.e, how is this experiment conducted and what
sort of equipment is required. Then some statement as to the proposed applicability
followed by the present contents would render this far more useful.
Thank you. Considering your feedback, a new section (2. PA devices - Noninvasive
imaging procedures) was added, after the introduction, which provides
an overview of current PA devices, typical experimental techniques, and figures.

Reviewer 2 Report

The article Photoacoustic Imaging Probes based on tetrapyrroles and related compounds is interesting since it explains current state of the art in an comprehensive way.

Comments:

The only comment is to the first sentence of Chapter 2. Basics of PA probes- the known strategies: the meaning of the sentence is not clear, so please rewrite.

Author Response

Thank you.

The heading was changed to: Basics of PA Probes - The known design strategies